# SemMAE: Semantic-Guided Masking for Learning Masked Autoencoders

**Gang Li**[1,2]*, **Heliang Zheng**[3], **Daqing Liu**[3], **Chaoyue Wang**[3], **Bing Su**[4], **Changwen Zheng**[1]†
Institute of Software, Chinese Academy of Sciences[1],
University of Chinese Academy of Sciences[2],
JD Explore Academy[3], Renmin University of China[4]
`ucasligang@gmail.com`, `{zhengheliang,liudaqing1,wangchaoyue9}@jd.com`,
`bingsu@ruc.edu.cn`, `changwen@iscas.ac.cn`

## Abstract

Recently, significant progress has been made in masked image modeling to catch up to masked language modeling. However, unlike words in NLP, the lack of semantic decomposition of images still makes masked autoencoding (MAE) different between vision and language. In this paper, we explore a potential visual analogue of words, i.e., semantic parts, and we integrate semantic information into the training process of MAE by proposing a Semantic-Guided Masking strategy. Compared to widely adopted random masking, our masking strategy can gradually guide the network to learn various information, i.e., from intra-part patterns to inter-part relations. In particular, we achieve this in two steps. 1) Semantic part learning: we design a self-supervised part learning method to obtain semantic parts by leveraging and refining the multi-head attention of a ViT-based encoder. 2) Semantic-guided MAE (SemMAE) training: we design a masking strategy that varies from masking a portion of patches in each part to masking a portion of (whole) parts in an image. Extensive experiments on various vision tasks show that SemMAE can learn better image representation by integrating semantic information. In particular, SemMAE achieves 84.5% fine-tuning accuracy on ImageNet-1k, which outperforms the vanilla MAE by 1.4%. In the semantic segmentation and fine-grained recognition tasks, SemMAE also brings significant improvements and yields the state-of-the-art performance. Our code is available at `https://github.com/ucasligang/SemMAE`.

## 1 Introduction

Together with transformers, masked language modeling (MLM) has revolutionized the field of self-supervised learning (SSL) in natural language processing (NLP), which enables training of generalizable NLP models containing over one hundred billion parameters [5]. The concept of MLM is quite intuitive, i.e., a portion of the data is removed and a model is trained to predict the removed content. Recently, significant progress has been made in masked image modeling to catch up to masked language modeling, where the masking mechanism is a key factor. Context encoder [28], an inpainting-based masked image modeling (MIM) pioneer, proposes to use a random and fix-shaped mask; SiT [2] and BEiT [3] use random "blockwise" masks, where patches in the local neighbourhood are masked together (also called GMML: group mask model learning); MAE [19] randomly masks out 75% patches of an image. Actually, masking mechanisms define the specific pretext task, i.e., what kind of information is to be exploited and what kind of information is to be predicted. Thus

---

*This work was performed when Gang Li was visiting JD Explore Academy as a research intern.
†Corresponding author.

36th Conference on Neural Information Processing Systems (NeurIPS 2022).

AttMask [21] studies the problem of which tokens to mask and proposes an attention-guided mask strategy to make informed decisions. ADIOS [30] takes one step further to "learn to mask" by adversarial training.

Although promising performance has been achieved, there is still a large gap for masked autoencoding (MAE) between vision and language due to different signal natures. A sentence can be semantically decomposed into words, while the semantic decomposition of an image is not trivial to be obtained. To find a visual analogue of words, we investigate part-based image representation. Specifically, the real world is composed of objects, which consist of different parts. Therefore, part-based image representation is a fundamental image representation method that fits the inherent properties of objects [8, 15, 16, 18, 20]. For example, part-based Pictorial Stracture [16] dominated the image representation field for several years in the early days of computer vision, and Deformable Part Model (DPM) [15] was also a milestone in image recognition and detection. Moreover, GLOM [20] argues that the hierarchical representation with five levels (i.e., the lowest level, sub-part level, part level, object level, and scene level) would be a powerful image representation method in the future. To this end, we argue that *semantic parts would be a potential visual analogue of words*. With such visual analogue, *more controllable hints* can be built up to guide the learning of MAE, thus high-level visual representations can be well learned.

In this paper, we first propose a self-supervised semantic part learning method to obtain semantic parts for each image. Our insight is that the spatial information to reconstruct an image is highly correlated to the position of semantic parts. In particular, our part learning model consists of a ViT-based encoder together with an attention module that generates a class token and multiple attention maps, and a StyleGAN-based decoder that reconstructs the original image. The attention maps are optimized to provide spatial information, and the class token is integrated into the decoder via AdaIN to provide texture information. We find that the optimized attention maps can indicate part positions, and we conduct an argmax operation to obtain part segmentation maps. After that, we study how semantic parts can facilitate the learning of MAE. We design a masking strategy that varies from masking a portion of patches in each part to masking a portion of (whole) parts in an image. Such a design can gradually guide the network to learn various information, i.e., from intra-part patterns to inter-part relations. Extensive experiments on various vision tasks (e.g., linear probing, fine-tuning, semantic segmentation, and fine-grained recognition) show that SemMAE can learn better image representation by integrating semantics.

Our contributions include 1) designing a self-supervised semantic part learning method that can generate promising semantic parts on multi-class datasets, i.e., ImageNet, and 2) verifying that semantic parts can facilitate the learning of MAE by proposing a semantic-guided masking strategy. While more importantly, we hope our attempts can provide insights for the community to study the visual analogue of words and unified vision and language modeling.

## 2  Related work

**Semantic part learning.** Part-based image representation is a fundamental image representation method that fits the inherent properties of objects [8, 15, 16, 18, 20]. However, due to the tremendous cost of labeling parts, there are still no large-scale datasets containing part labels. Thus previous works are mainly two-fold, i.e., unsupervised/weakly-supervised part learning and few-shot part segmentation. Unsupervised/weakly-supervised part learning methods [10, 22, 39] propose to mine part information by leveraging spatial priors, the semantics of convolutional channels, or designing contrastive proxy tasks. Few-shot part segmentation methods [4, 29, 38] mainly learn an additional classifier over pre-trained features that are trained by GAN, self-supervised contrastive learning, or denoising diffusion probabilistic modeling. Although promising results have been obtained, these models are designed to deal with fine-grained datasets, where all images belong to a single super-class (e.g., birds, cars, or human faces). It is much more challenging to solve the problem of unsupervised part learning on multi-class datasets such as ImageNet. With the development of ViT and self-supervised learning (SSL), some recent works show a potential solution. In particular, DINO [6] and iBOT [42] have observed intuitive semantics in the ViT trained by their SSL methods, where the multi-head attention maps can somehow indicate different semantic parts of an object. Inspired by these works, we design a reconstruction-based method to further refine the attention maps learned by iBOT to obtain semantic parts on the ImageNet dataset.

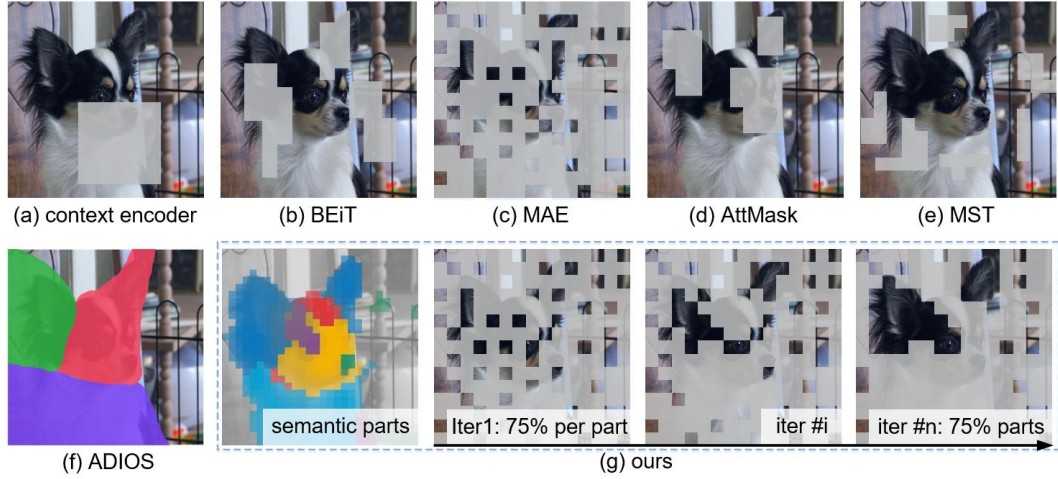

(a) context encoder    (b) BEiT    (c) MAE    (d) AttMask    (e) MST

(f) ADIOS      semantic parts    Iter1: 75% per part    iter #i    iter #n: 75% parts    (g) ours

**Figure 1:** Comparison of different masking strategies. Detailed information for each compared model can be found in Section 2 Masked Image Modeling.

**Masked image modeling.** Inspired by the success of Masked Language Modeling (MLM) [5, 12] in pre-training of the NLP field, Masked Image Modeling (MIM) has been proposed recently and exhibits promising potential for visual pre-training [3, 7, 19, 30]. Existing works mainly study the problem of MIM from two directions, i.e., regression targets and masking strategies. In terms of regression targets, BeiT [3], mc-BEiT [24], and PeCo [13] adopt tokens produced by VQ-VAE [32] or its variants. MaskFeat [35] studies a broad spectrum of feature types and proposes to regress Histograms of Oriented Gradients (HOG) features of the masked content. MAE [19] and SimMIM [37] argue that predicting RGB values of raw pixels by direct regression performs no worse than the patch classification approaches with complex designs. In this paper, we follow MAE [19] to adopt the most simple and intuitive raw pixels regression. In terms of masking strategies, SiT [2], MC-SSL0.0 [1] and BeiT [3] use a block-wise masking strategy, where a block of neighbouring tokens arranged spatially are masked. MAE [19] and SimMIM [37] use random masking with a large masked patch size or a large proportion of masked patches. MST [25] and AttMask [21] propose to use attention maps to guide the masking strategy, where the former proposes to mask the nonessential regions to preserve crucial patches while the latter proposes to learn image representations with challenging tasks by masking the most attended tokens. Moreover, ADIOS [30] proposes to learn an optimal mask by adversarial training. Compared to these works, our SemMAE takes one step further and explicitly learn semantic parts to build reasonable hints for masked image modeling. Figure 1 is an illustration of different masking strategies.

## 3 Semantic-guided masked autoencoders

We propose a Semantic-guided Masked Autoencoder (SemMAE) for self-supervised image representation learning with mask image modeling. The framework of SemMAE is shown in Figure 2, which consists of two key components, i.e., Semantic Part Learning (A) and Semantic-Guided Masking (B). First, given an image in Figure 2 (a), we extract the class token in Figure 2 (b) and patch tokens in Figure 2 (c) by an iBOT-pretrained ViT. After that, we learn an embedding over the class token to obtain part tokens in Figure 2 (d). We calculate the correlation of each part token to patch tokens to obtain attention maps in Figure 2 (e), whose texture information is further removed by a large-kernel blur operation. The attention maps are optimized by a diversity constraint and a reconstruction task where the attention maps and the class token are fed into a StyleGAN-based decoder to control the spatial and texture information of the reconstructed image, respectively. Finally, we conduct argmax over the attention maps to obtain part segmentation maps in Figure 2 (f) and used them to guide the mask generation for MAE. Specifically, we design a masking strategy that varies from masking a portion of patches in each part to masking a portion of (whole) parts in an image. Such a design can gradually guide the network to learn various information, i.e., from intra-part patterns to inter-part relations.

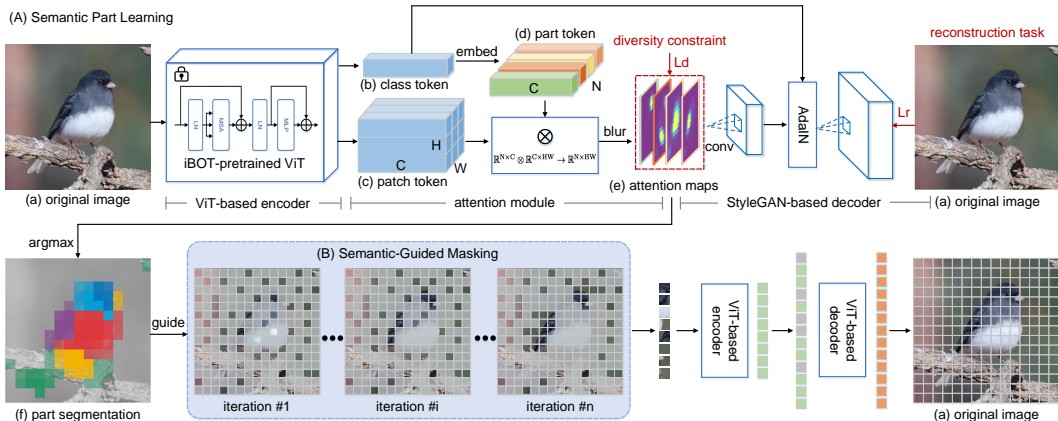

**Figure 2:** An illustration of the proposed SemMAE. (A) Semantic Part Learning. A ViT-based encoder takes as input an image in (a) and produces a class token in (b) and patch tokens in (c). Our attention module first learns to embed the class token into part tokens in (d) and then generates an attention map for each part token by calculating the correlation between the part token and patch tokens. As an objective function of the attention maps, our StyleGAN-based decoder learns to reconstruct the original image from attention maps with texture information from the class token. (B) Semantic-Guided Masking. We conduct argmax over the attention maps to obtain part segmentations in (f), which are used to guide the mask generation. During the training of the MAE, the masks vary from a portion of patches in each part to a portion of (whole) parts in an image.

## 3.1   Semantic part learning

In this subsection, we introduce our self-supervised semantic part learning method. Previous unsupervised/weakly-supervised part learning methods are mainly designed to deal with single-class datasets (i.e., fine-grained datasets where images belong to the same superclass). Few methods are able to solve this problem under a multi-class scenario (e.g., ImageNet). While some recent works (i.e., DINO [6] and iBOT [42]) on ViT-based self-supervised learning show that the multi-head attention maps in their model can somehow indicate different semantic parts of an object. In this work, we take advantage of semantics learned in iBOT and design a reconstruction task together with a diversity constraint to refine and obtain semantic parts.

In particular, given an image $\mathbf{I}$, we first use an iBOT-pretrained ViT to extract its features, i.e., a class token $\mathbf{F}_{\mathrm{c}} \in \mathbb{R}^{C \times 1}$ and patch tokens $\mathbf{F} \in \mathbb{R}^{C \times HW}$. Then, we embed the class token into $N$ part tokens $\mathbf{F}_{\mathrm{p}} \in \mathbb{R}^{C \times N}$. The main idea of such embedding is to re-weight feature channels of the class token. As shown in previous methods [39], feature channels may be corresponding to specific semantics and channel re-weighting can group channels with similar semantics together to obtain semantic part features. Thus we can obtain part tokens by:

$$\mathbf{F}_{\mathrm{p}}^{(i)} = \mathbf{F}_{\mathrm{c}} \circ \mathrm{sigmoid}(\mathbf{W}_{\mathrm{c2}}^{(i)} \tanh(\mathbf{W}_{\mathrm{c1}}^{(i)} \mathbf{F}_{\mathrm{c}})), \tag{1}$$

where $i \in [1, 2, ..., N]$, $\mathbf{F}_{\mathrm{p}}^{(i)} \in \mathbb{R}^{C \times 1}$ is the $i^{th}$ column vector of $\mathbf{F}_{\mathrm{p}} \in \mathbb{R}^{C \times N}$, $\circ$ indicates hadamard product, $\mathbf{W}_{\mathrm{c1}}^{(i)}$ and $\mathbf{W}_{\mathrm{c2}}^{(i)}$ are embedding weights, $\tanh(\cdot)$ and $\mathrm{sigmoid}(\cdot)$ are activation functions.

After that, we calculate the correlation of each part token to the patch token in each position, thus we can obtain attention maps, i.e., the possibility of a semantic part to appear in each position:

$$\mathbf{M} = \mathbf{F}_{\mathrm{p}} \otimes \mathbf{F} := \mathrm{softmax}(\mathbf{F}_{\mathrm{p}}^{T} \mathbf{W}_{\mathrm{p}}^{T} \mathbf{W} \mathbf{F}), \tag{2}$$

where $\mathbf{M} \in \mathbb{R}^{N \times HW}$ denotes $N$ attention maps, $\otimes$ indicates correlation function, which is implemented by $\mathrm{softmax}(\mathbf{F}_{\mathrm{p}}^{T} \mathbf{W}_{\mathrm{p}}^{T} \mathbf{W} \mathbf{F})$ in our work. $\mathbf{W}_{\mathrm{p}}$ and $\mathbf{W}$ are embedding matrixes.

To learn such multi-attention maps (i.e., to optimize the parameters in Equation 1 and Equation 2), we propose a reconstruction task. Our insight is that the spatial information to reconstruct an image is highly correlated to the position of semantic parts. Thus, we adopt a StyleGAN-based decoder to reconstruct the original image based on the spatial information from the attention maps and the texture information from the class token. To ensure the attention maps learn spatial information, we 1) remove texture information from the attention maps by conducting a large-kernel blur operation

and 2) further feed the blurred attention maps to stacked convolutional layers. To integrate the texture information from the class token into the decoder, we use Adaptive Instance Normalization (AdaIN) operation, which is widely used to integrate texture/style information:

$$[\mathbf{F}_{\mathrm{d}}]_i = \mathrm{AdaIN}([\mathrm{conv}(\mathbf{M})]_i, \mathbf{F}_{\mathrm{c}}) \coloneqq [\mathbf{W}_{\mathrm{s}}\mathbf{F}_{\mathrm{c}}]_i \frac{[\mathrm{conv}(\mathbf{M})]_i - \mu([\mathrm{conv}(\mathbf{M})]_i)}{\sigma([\mathrm{conv}(\mathbf{M})]_i)} + [\mathbf{W}_{\mathrm{b}}\mathbf{F}_{\mathrm{c}}]_i, \quad (3)$$

where each feature channel $[\mathrm{conv}(\mathbf{M})]_i$ is normalized separately, and then scaled and biased using the corresponding scalar components from the embedded class token $\mathbf{F}_{\mathrm{c}}$. $\mathbf{F}_{\mathrm{d}}$ denotes the convolutional feature in the decoder, $\mathbf{W}_{\mathrm{s}}$ and $\mathbf{W}_{\mathrm{b}}$ are embedding weights, $[\cdot]_i$ denotes the $i^{th}$ feature channel, $\mathrm{conv}(\cdot)$ denotes convolutional layers, $\mu(\cdot)$ and $\sigma(\cdot)$ calculate the mean and variance values, respectively. The reconstructed image $\hat{\mathbf{I}}$ can be obtained by stacking convolutional and AdaIN layers:

$$\hat{\mathbf{I}} = \mathrm{conv}(\mathrm{AdaIN}(\mathrm{conv}(\mathbf{F}_{\mathrm{d}}), \mathbf{F}_{\mathrm{c}})). \tag{4}$$

We use the Mean squared error (MSE) loss function to optimize such reconstruction task:

$$\mathcal{L}_{rec}(\mathbf{I}, \hat{\mathbf{I}}) = \frac{1}{HW} \sum_{i,j}^{HW} (\mathbf{I}(i,j) - \hat{\mathbf{I}}(i,j))^2. \tag{5}$$

Moreover, to obtain diverse multiple attention maps, we follow previous work [40] and add a diversity constraint over attention maps:

$$\mathcal{L}_{div}(\mathbf{M}) = \frac{1}{N^2}\left(\sum_{i \neq j} (0 - \frac{\mathbf{m}_i \mathbf{m}_j^T}{\|\mathbf{m}_i\|_2 \|\mathbf{m}_j\|_2})^2 + \sum_{i=j} (1 - \frac{\mathbf{m}_i \mathbf{m}_j^T}{\|\mathbf{m}_i\|_2 \|\mathbf{m}_j\|_2})^2\right), \tag{6}$$

where attention maps are optimized to be different from each other, $\mathbf{m}_i$ and $\mathbf{m}_j$ denotes the $i^{th}$ and $j^{th}$ attention map, respectively. The overall objective function can be denoted by:

$$\mathcal{L} = \mathcal{L}_{rec}(\mathbf{I}, \hat{\mathbf{I}}) + \lambda \mathcal{L}_{div}(\mathbf{M}), \tag{7}$$

where $\lambda$ is the loss weight.

## 3.2   Semantic-guided masking

After finished semantic part learning, we move to the next stage, i.e., semantic-guided MAE training. Our informed masking strategy is based on the part information learned in Subsection 3.1. Specifically, we can obtain multiple attention maps by Equation 2, where each attention map $\mathbf{m} \in \mathbb{R}^{H \times W}$ indicates the possibility of the corresponding semantic part appearing in $H \times W$ positions. Thus we conduct $\mathrm{argmax}(\cdot)$ operation over attention maps to obtain part segmentation, where each patch is classified into a particular semantic part. The patches in the same semantic part compose a visual analogue of words, which are semantically meaningful. To leverage such visual analogue of words for MAE training, a most intuitive way is to mask a portion of semantic parts and learn to predict the masked semantic parts by other parts. However, due to the learned semantic parts being coarse-grained (e.g., 6 parts for each image), we experimentally find that such a masking strategy makes the task too hard to effectively learn image representations.

To this end, we propose an easy-to-hard reconstruction task, which can provide reasonable hints (i.e., visible patches) for the model to predict the masked patches during the training process of the MAE. Specifically, at the beginning of the training process, we mask a portion of patches in each part, thus the masked patches can be predicted based on the visual patches that belong to the same semantic part. Such a design can facilitate the models to learn intra-part patterns. After that, we gradually mask all patches belonging to some parts and a portion of patches belong to the remaining parts. Finally, we mask all patches belonging to a portion of parts and predict the remaining patches belong to the other parts, where inter-part relations or visual reasoning ability can be learned.

Algorithm 1 shows the details to obtain the number of masked patches for each semantic part. First, we define two masking settings, i.e., 1) mask a portion of patches in each part and 2) random select some parts to mask (the whole part). The number of masked patches for each semantic part can be calculated for these two settings. After that, we introduce an interpolation hyper-parameter $\alpha$. A small $\alpha$ means the first setting dominates the masking strategy, and vice versa. $\alpha$ is adjusted based on training iterations and keeps increasing during the training process. Finally, we random mask a certain number of patches based on the calculated masking number.

---

**Algorithm 1** Algorithm of Semantic-Guided Masking in a PyTorch-like style.

---

**Input:** $L, x, num\_patches, mask\_ratio, total\_epoches, epoch$

  \# $L$: the number of patches per image.
  \# $x \in \mathbb{R}^{L \times C}$: the token embeddings of image patches.
  \# $num\_patches \in \mathbb{R}^{N \times 1}$: the patch number of each part, where $N$ is the number of parts.
  \# $mask\_ratio$: the ratio of masked patches.
  \# $total\_epoches$: the number of pre-training epochs.
  \# $epoch$: current epoch number.

  \# mask a portion of patches in each part
 1: $num\_mask1 = mask\_ratio * num\_patches$
  \# randomly select some parts to mask (with tricks to ensure a fixed mask ratio)
 2: $shuffle\_num\_patches = shuffle\_parts(num\_patches)$
 3: $marks = \text{L} * mask\_ratio\text{-cumsum}(shuffle\_num\_patches) + shuffle\_num\_patches$
 4: $marks\_remains = \text{where}(marks < 0, 0, marks)$
 5: $num\_mask2 = \text{where}(marks\_remains < shuffle\_num\_patches, marks\_remains,$
   $shuffle\_num\_patches)$
 6: $num\_mask2 = unshuffle\_parts(num\_mask2)$
  \# adaptive masking by interpolating between num_mask1 and num_mask2
 7: $\alpha = \left(\frac{epoch}{total\_epoches}\right)^{\gamma}$
 8: $num\_mask = (1 - \alpha) * num\_mask1 + \alpha * num\_mask2$
 9: **return** $num\_mask$ \# $num\_mask \in \mathbb{R}^{N \times 1}$: the number of patches to be masked in each part.

---

## 4 Experiments

### 4.1 Experiment setup

**Semantic part learning.** As introduced in Section 3.1, we use ViT-small [14] as our part learning encoder, which is pre-trained by a self-supervised method iBOT [42]. We follow iBOT [42] to learn 6 semantic parts for each image, as the head number of the multi-head attention in ViT-Small is 6. The size of the blur kernel is experimentally set to be 7, and the loss weight $\lambda$ in Equation 7 is set to be 0.03. The experiment is performed on ImageNet-1k [11] dataset. The parameters of the ViT-based encoder are fixed, and we only optimize the attention module and the StyleGAN-based decoder. Our model converges fast, which only takes 2 hours on one A100 GPU card.

**Semantic-guided MAE training.** We follow MAE [19] and adopt an encoder-decoder structure to perform MIM. Our method is general for ViT backbones, while most experiments are conducted with a relatively small version, i.e., the original ViT-Base [14], due to the limitation of computation resources. We follow the most comment setting to optimize our model by AdamW [27] with a learning rate of 2.4e-3. The batch size is set to be 4096, and the weight decay is set to be 0.05. We use a cosine learning rate strategy [26] with warmup [17]. The warmup number is set to be 40 epochs, and we pre-train our model for 800 epochs. For data augmentation, we only employ random horizontal flipping in our pre-training stage. The hyper-parameter $\gamma$ in Algorithm 1 is experimentally set to be 2. Our model is trained on 16 A-100 GPUs for 3 days, and more details can be found in our code, which is in the supplemental material and will be made publicly released.

Table 1: Quantitative evaluation of the effectiveness of integrating semantic information for MAE.

| Setting | 16×16 patch size | | 8 × 8 patch size | |
|---|---|---|---|---|
| | MAE [19] | SemMAE | MAE [19] | SemMAE |
| Linear probing | 63.7 | **65.0** | 66.8 | **68.7** |

### 4.2 Semantic-guided MAE

**The effectiveness of integrating semantic information.** We conduct experiments under two different settings (i.e., with a patch size of 16×16 and 8×8) to verify the effectiveness of integrating semantic information for training MAE. The results in Table 1 show that integrating semantic information can bring 1.3% and 1.9% accuracy gains for linear probing, respectively. As we use 8×8 patch size to

**Table 2:** The optimal patch size for different models.

| Model | Patch size | Fine-tuning Acc.(%) | Model | Patch size | Fine-tuning Acc.(%) |
|---|---|---|---|---|---|
| SimMIM [37] | 32x32 | **82.8** | MAE [19] | 16x16 | **83.26** |
| | 16x16 | 82.7 | | 8x8 | 83.10 |
| | 8x8 | 82.1 | SemMAE | 16x16 | 83.34 |
| | 4x4 | 82.0 | | 8x8 | **84.50** |

learn semantic parts, the coarse-grained patch (i.e., large patch size) in the pre-training stage would cause imprecise part segment and suppresses the benefits of semantic parts. To further study the impact of patch size for masked image modeling, we conduct fine-tuning experiments in Table 2. It can be observed that in SimMIM and original MAE, a larger patch size performs better; while in our SemMAE, more precise semantic parts with 8×8 patch size can significantly improve the performance. Thus in the following experiments, we adopt 8×8 patch size for SemMAE. It is notable that although using a smaller patch size, our parameters and computational cost do not increase during pre-training and linear probing as only 1/4 patches in each image are used. Specifically, we leverage the learned attentions maps to remove 3/4 patches that are most likely to be the background.

**A detailed study on masking strategies.** Once obtained semantic parts, a most intuitive way to leverage such visual analogue of words for MAE training is to mask a portion of semantic parts and make the model to predict the removed content. However, due to the learned semantic parts being coarse-grained (e.g., 6 parts for each image), we experimentally find that such a masking strategy makes the task too hard to effectively learn image representations. The results can be found in Table 3, where masking 75% parts cause 13.9% performance drops compared to random masking. Moreover, it can be observed that masking 75% patches per part achieves comparable results with random masking. The self-supervised learning task of masking 75% patches per part would encourage the model to learn local contexts/intra-part patterns, and masking 75% parts would encourage the model to learn inter-part relations. Interestingly, we find that the former task can enable the model to further learn better image representation in the latter task. The results in Table 3 show that our proposed adaptive masking strategy (i.e., varying from masking 75% patches per part to masking 75% parts gradually) with $\gamma = 2$ yields the best performance.

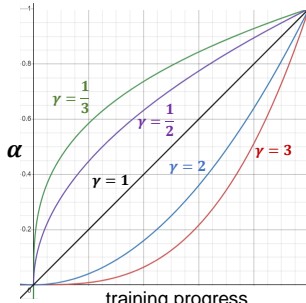

**Figure 3:** The curves of $\alpha$ and $\gamma$.

| Mask strategy | $\alpha$ | $\gamma$ | Linear probing |
|---|---|---|---|
| Random masking | – | – | 66.8 |
| Mask 75% patches | 0 | – | 66.5 |
| Mask 75% parts | 1 | – | 52.9 |
| Adaptive masking | $1 \rightarrow 0$ | 1 | 63.3 |
| | $0 \rightarrow 1$ | 1/3 | 66.2 |
| | | 1/2 | 67.3 |
| | | 1 | 67.9 |
| | | 2 | **68.7** |
| | | 3 | 68.6 |

**Table 3:** Quantitative evaluation of different masking strategies.

### 4.3 Semantic part learning

We first evaluate the effectiveness of our proposed semantic part learning method. Both qualitative and quantitative experiments are conducted. Note that most of the previous part learning models are designed for single-class datasets and cannot be effectively applied to ImageNet. iBOT [42] is not proposed for part learning, while the multi-head attention in their model achieves the state-of-the-art part learning performance on ImageNet. Figure 4 shows the qualitative comparison of our model and iBOT [42], and it can be observed that our model can generate more complete semantic part segmentation maps where different parts and the background are better separated with less noise. Moreover, as there is no part segmentation ground truth, we conduct quantitatively evaluation in an indirect way by training SemMAE and analyzing ImageNet classification performance. The results in Table 4 show that the semantic parts obtained by iBOT are not able to benefit the learning of MAE, while our semantic part learning methods can generate more precise part segmentation maps, which are vital to learning a better image representation.

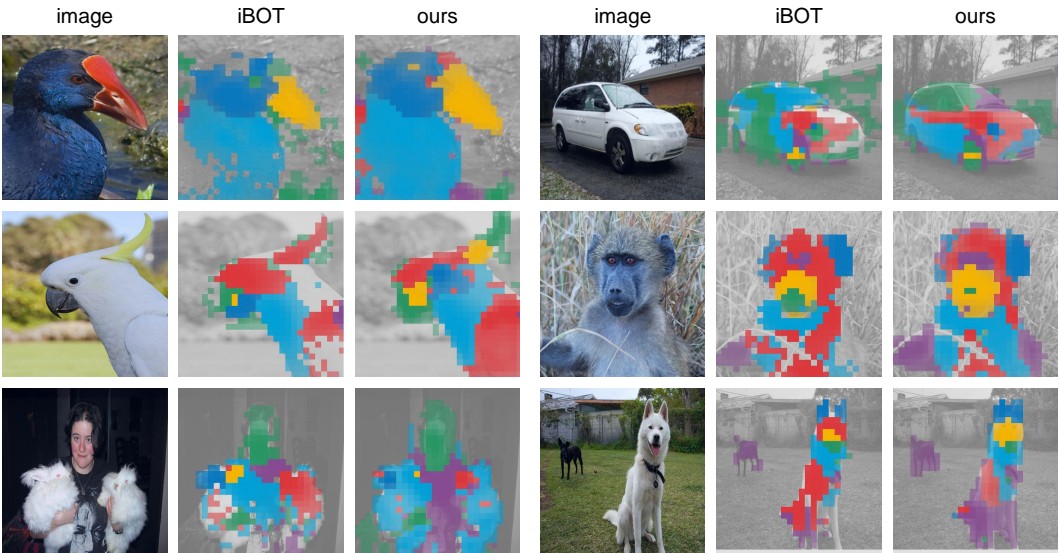

**Figure 4:** Qualitative comparison of semantic part learning. Different color indicates different semantic parts, and it can be observed that our model can better separate different parts and the background with less noise.

**Table 4:** Quantitative evaluation of semantic part learning in terms of classification accuracy(%).

| Semantic parts for masking | Baseline (w/o parts) | iBOT-initialized parts | Our learned parts |
| --- | --- | --- | --- |
| Linear probing Acc. (%) | 63.7 | 63.6 | **65.0** |

**Table 5:** System-level comparison on ImageNet-1k in terms of classification accuracy using ViT-Base as the encoder. Note that we list the best performance in previous papers with $224 \times 224$ inputs, and some experiment settings (e.g., training epochs and patch size) may be different.

| Method | Pre-train dataset | Pre-train epoches | Linear probing | Fintuning |
| --- | --- | --- | --- | --- |
| *Traning from scratch* | | | | |
| ViT$_{384}$ [14] | - | - | - | 77.9 |
| DeiT [31] | - | - | - | 81.8 |
| ViT [19] | - | - | - | 82.3 |
| *Contrastive-based SSL Pre-Training* | | | | |
| AttMask [42] | ImageNet-1K | 100 | 75.7 | - |
| DINO [6] | ImageNet-1K | 300 | 78.2 | 82.8 |
| MoCo v3 [9] | ImageNet-1K | 300 | 76.5 | 83.2 |
| iBOT [42] | ImageNet-1K | 1600 | 79.5 | 84.0 |
| *MIM-based SSL Pre-Training* | | | | |
| BeiT [3] | ImageNet-1K | 800 | 56.7 | 83.2 |
| MAE [19] | ImageNet-1K | 1600 | 68.0 | 83.6 |
| SimMIM [37] | ImageNet-1K | 800 | 56.7 | 83.8 |
| SemMAE | ImageNet-1K | 800 | **68.7** | **84.5** |

## 4.4 Compared with other methods on ImageNet

Linear probing and fine-tuning on ImageNet-1K classification dataset is the most common setting to evaluate SSL methods. We collect all competitive methods that report their results on ImageNet-1K dataset. For example, we do not include the related work MST [25] and ADIOS [30] as they evaluate their model on other benchmarks. Table 5 shows the comparison of our model and previous models in terms of linear probing and fine-tuning. For a fair comparison, all experiments adopt the same input size, i.e., $224 \times 224$ unless specified otherwise. Compared with "training from scratch", our SemMAE can significantly improve the performance for both linear probing and fine-tuning. For linear probing, our SemMAE outperforms the most competitive MIM-based methods by 0.8%

even with fewer training epochs. For fine-tuning, our SemMAE achieves 84.5% top-1 classification accuracy, outperforming SimMIM[37] and MAE[19] by 0.9% and 0.7% respectively. Moreover, our SemMAE can also surpass previous contrastive learning-based methods[6, 9] for fine-tuning.

## 4.5 Downstream tasks

**Fine-grained image classification.** Table 6 shows our results of transfer learning on fine-grained datasets. Our model can surpass the most competitive MAE [19] with a clear margin, i.e., 0.3%, 0.6%, and 0.2% on the iNaturalists[33], CUB-Bird[34], and Stanford-Cars[23] dataset, respectively. These results show the promising transfer ability of our SemMAE for downstream classification tasks.

**Table 6:** Fine-tuning results on fine-grained datasets.

| Method | $iNa_{19}$ | CUB | Cars |
|---|---|---|---|
| BeiT [3] | 79.2 | - | 94.2 |
| DINO [3] | 78.6 | - | 93.0 |
| iBoT [19] | 79.6 | - | 94.3 |
| MAE [19] | 81.8 | 86.5 | 94.2 |
| SemMAE | **82.1** | **87.1** | **94.4** |

**Semantic segmentation.** Semantic segmentation aims to assign a label to each pixel of the input image. We evaluate our SemMAE on the widely used semantic segmentation dataset ADE20K [41], which contains 25K images and 150 semantic categories. We follow the most common setting to use the task layer in UPerNet [36] and fine-tune the pre-trained ViT-Base model. We use the standard setting that pre-train a ViT-Base model with a patch size of $16 \times 16$ and fine-tunes 160K iterations with a batch size of 16. Such an experiment can validate the transfer ability of our SemMAE for semantic segmentation. As shown in Table 7, SeMAE surpasses MAE by 0.2 (46.3 vs. 46.1) mIoU and outperforms the supervised pre-train model by 1.0 mIoU. These results show the promising transfer ability of our SemMAE for dense prediction visual tasks.

**Table 7:** Semantic segmentation results on ADE-20K.

| Method | mIoU |
|---|---|
| Supervised Pre-Training | 45.3 |
| *Self-Supervised Pre-Training* | |
| BeiT | 45.8 |
| MAE(800 epochs) | 46.1 |
| SemMAE (Ours) | **46.3** |

## 5 Conclusion

In this paper, we study the visual analogue of words and propose a semantic-guided masked autoencoder model to reduce the gap between masked language modeling and masked image modeling. Our proposed self-supervised semantic part learning method can generate promising semantic parts on ImageNet and we show that the learned semantic parts can facilitate the learning of MAE. Unlike the main-stream random masking strategy, our semantic-guided mask strategy can effectively integrate semantic information in the pre-training process. Extensive experiments with superior results show the effectiveness of our SemMAE.

**Limitations**: due to the lack of part segmentation labels, the semantic part in our work is kind of coarse (e.g., 6 parts per image), making it not an ideal visual analogue of words yet. Moreover, using a small patch size increases the computational cost in the fine-tuning stage. In the future, we will 1) investigate finer-grained semantic parts (e.g., 20-30 parts per image) by few-shot part segmentation and 2) replace the widely obtained patch-based tokenization with part-based tokenization to further reduce the gap between vision and language modeling.

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
