# OpenReview forum: "SemMAE: Semantic-Guided Masking for Learning Masked Autoencoders"
_NeurIPS.cc/2022/Conference — NeurIPS 2022 Accept_

### Official Review · Reviewer_YRge · 2022-07-11

**Rating:** 7
**Confidence:** 4
**Soundness:** 4 excellent
**Presentation:** 3 good
**Contribution:** 4 excellent

**Summary:**

This paper proposed a new semantic-guided masking strategy for unsupervised vision transformer training. The paper claims that semantic parts of objects are the visual analogue of words, masking semantic parts from intra-part patterns to inter-part would help the vision transformer learn better representations. For such purpose, the paper first designs a self-supervised semantic part learning method that aims to generate semantic parts of images from the ImageNet dataset. Then, the proposed semantic-guided masking strategy performs the unsupervised model training. Most of the experiments were performed on the ViT-base backbone and showed improvements on unsupervised representation learning.

**Questions:**

1.For the proposed SemMAE, decreasing patch size largely improved its performance. Please explain the reason behind this and compare it to the opposite behavior in both SimMIM and the original MAE.

2.Why six parts per image were selected in the paper? For different kinds of objects in the ImageNet dataset, what are the six parts corresponded to?

**Limitations:**

Yes, the authors adequately discussed the limitations of this paper. Meanwhile, potential negative societal impacts are not found yet.

**Strengths And Weaknesses:**

[Strengths]
1.This paper is well organized. It raises an interesting and reasonable assumption, i.e., semantic parts in an image carry important semantic information and may benefit unsupervised masked training strategy. To explore the difference between language tokens and image tokens seems a meaningful topic and this paper seems to find a new way to define semantic tokens for vision transformers.

2.If the ground-truth semantic part segmentations are given, the comparisons with existing MAE methods should be unfair. Thus, the paper designs a self-supervised semantic parts prediction model. We can see that the semantic part segmentation results would largely influence the masked learning results. It seems the proposed semantic part learning method works well in exploring parts' information that is contained in the representations extracted from a pre-trained transformer.

3.The experimental settings are reasonable and related results are promising. Most claims in this paper can be supported by their experiments.

[Weaknesses]
1.Besides objects (or semantic parts), the background is an important and large part contained in images. This paper does not discuss how to deal with the background. If we regard the semantic parts as the visual analog of words, what is the background for? How should we treat it?

2.I am wondering why this paper chooses the masking training strategy to validate its idea (i.e., semantic parts of objects are the visual analogue of words). Are there any other tasks related to this idea? In addition, what is the relationship between the idea and the masking strategy (i.e., masking parts from intra-part patterns to inter-part)? These questions were not well explained?

---

> ### Author Response · Authors · 2022-08-02
> **Response to Reviewer YRge**
>
> Thank you for your review. We appreciate the positive comments on "well organized", "interesting and reasonable assumption", and "reasonable settings and promising results". We will consider each of your questions or concerns below.
>
> #### Q1: Besides objects (or semantic parts), the background is an important and large part contained in images. This paper does not discuss how to deal with the background. If we regard the semantic parts as the visual analog of words, what is the background for? How should we treat it?
>
> A1: In this paper, we adopt two ways to deal with the background, i.e., 1) random masking 75% patches of the background and 2) ignoring the background to calculate the loss. Both of these settings show the effectiveness of our proposed SemMAE (Line 206-221 and Table 2). While we agree that it is still an open problem to find the visual analog of words for the background, and we will further study this problem in future work.
>
>
> #### Q2: I am wondering why this paper chooses the masking training strategy to validate its idea (i.e., semantic parts of objects are the visual analogue of words). Are there any other tasks related to this idea? In addition, what is the relationship between the idea and the masking strategy (i.e., masking parts from intra-part patterns to inter-part)? These questions were not well explained?
>
> A2: *The choice of the task:* the research on mask language modeling (MLM) inspires us to choose the task of mask image modeling (MIM) to explore the visual analogue of words. Specifically, the success of MLM shows that the high-level representations of texts can be learned by masking semantic words. Thus we explore the problem of masking semantic meaningful patches (i.e., visual analogue of words) in MIM.
>
> *The relationship between the idea and the masking strategy:* Once we obtained part segmentations, a most intuitive way to validate our idea is to mask a portion of semantic parts. However, as the learned semantic parts are coarse-grained (e.g., 6 parts for each image), it is too hard to directly reconstruct the masked parts. To this end, we design a masking strategy that can gradually guide the network to learn from intra-part patterns to inter-part relations (i.e., parts reconstruction).
>
> *Other related tasks:* There are several tasks that can be used to explore the visual analogue of words, e.g., multimodal understanding tasks and multimodal generation tasks. And we are interested in further discussing this problem in multimodal tasks.
>
> #### Q3: For the proposed SemMAE, decreasing patch size largely improved its performance. Please explain the reason behind this and compare it to the opposite behavior in both SimMIM and the original MAE.
>
> A3: For random masking methods (SimMIM and MAE), decreasing patch size makes the model focus on learning and reconstructing local correlations, thus long-range correlations and high-level representations can not be well obtained.
>
> While our SemMAE can get rid of such problems our masking strategy can guide the model to learn from local correlations to long-range correlations. Moreover, decreasing patch size makes part segmentations more precise.
>
>
> #### Q4: Why six parts per image were selected in the paper? For different kinds of objects in the ImageNet dataset, what are the six parts corresponded to?
>
> A4: The number of parts is a hyperparameter, which is experimentally set to be 6. Specifically, we tried more parts (e.g., 8 parts), while we found the segmentation results are kind of noisy; we also tried fewer parts (e.g., 4 parts), while we found the segmentation results are kind of coarse. Figure 4 shows several examples of the learned parts (best viewed in color). We will show more cases and list the corresponding parts in the revised version.

---

### Official Review · Reviewer_KRLr · 2022-07-11

**Rating:** 3
**Confidence:** 4
**Soundness:** 2 fair
**Presentation:** 2 fair
**Contribution:** 2 fair

**Summary:**

A semantic-guided masking modeling method is proposed in this paper. To achieve this, an iBOT-pretrained ViT is used to extract the features, and then StyleGAN-based decoder is trained to learn semantic parts. Finally, the learned semantic parts attention maps are used to guide the mask modeling per-training. Extensive experiments are performed on classification and semantic segmentation downstream tasks.

**Questions:**

- Please see *Strengths And Weaknesses*
- In the end of the line 155: convolutional **can** AdaIN layers --> convolutional **and** AdaIN layers
- How about the total training cost including the iBOT pre-training, part learning, semantic guide pre-training, compared to the other pre-training methods.

**Limitations:**

The proposed method assumes that the images are object-centric. It's true for ImageNet dataset. How about the proposed method on none-object images, e.g. texture images, scene images, and et.al?

**Strengths And Weaknesses:**

# Strengths
- Semantic-guide MAE training is an interesting idea. But it does not convice me that the reason is just to have a potential visual analogue of words.  I encourage the authors to make it clearer why we need semantic-guide MAE training, and more efficient.

# Weaknesses
- The proposed methods involved too many components, iBOT feature extractor, including StyleGAN-based decoder, an intermediate reconstruction tasks, gradually masking strategy, and et.al.  to make it work. The complicated whole framework makes it hard to analyze how the proposed method works.
- It lacks the intuitive explanation why the proposed intermediate reconstruction tasks can help to learn the part attention maps.
- The improvement on downstream tasks is marginal as shown in Table 6 and Table 7.

---

> ### Author Response · Authors · 2022-08-02
> **Response to Reviewer KRLr**
>
> Thanks for your constructive comments. We will consider each of your concerns below.
>
> #### Q1: [Strengths] Semantic-guide MAE training is an interesting idea. But it does not convice me that the reason is just to have a potential visual analogue of words. I encourage the authors to make it clearer why we need semantic-guide MAE training, and more efficient.
>
> A1: Besides finding potential visual analogue of words to reduce the gap between vision and language, semantic-guide MAE training can guide the model to learn from inter-part patterns to intra-part relations. Experimental results in Table 3-7 show the effectiveness of leveraging semantic information for representation learning. Thanks for your advice, and we will add more analyses of the motivation of semantic-guide MAE training to the following revised paper.
>
>
> #### Q2: The proposed methods involved too many components, iBOT feature extractor, including StyleGAN-based decoder, an intermediate reconstruction tasks, gradually masking strategy, and et.al. to make it work. The complicated whole framework makes it hard to analyze how the proposed method works.
>
> A2: It would be much clearer to analyze our work by two folds. 1) self-supervised parts learning, and 2) learning semantic-guided MAE. Although they are working together to explore the effectivenesses of parts in image understanding, they can be understood as two independent components which have their own contributions.
>
> First, to learn parts without any ground truth labels, we design a reconstruction task to refine the attention maps generated by iBOT. Inspired by the feature learning capacity of the style-based generator architecture, we creatively proposed a style-based decoder for better parts exploration. Experimental results in Table 4 (main submission) show the effectiveness of our refinement, i.e., the proposed self-supervised parts learning strategy.
>
> Second, we proposed a simple yet effective mask training strategy, i.e., semantic-guided MAE. Our experimental results in Table 1 (main submission) validate our motivation that introducing part information will benefit image modeling. We can reasonably assume that the performance of image modeling will be further improved if better part information can be learned in the future.
>
> #### Q3: It lacks the intuitive explanation why the proposed intermediate reconstruction tasks can help to learn the part attention maps.
>
> A3: The reconstruction task can encourage spatial/position consistency between the attention maps and original images [10]. Specifically, as shown in dense prediction tasks and image translation tasks, stacked convolutions tend to preserve the spatial/position information of the inputs. Thus we conduct stacked convolutions over attention maps for reconstruction. And we experimentally find that the spatial information to reconstruct an image is highly correlated to the position of semantic parts.
>
> [10] Choudhury, S., Laina, I., Rupprecht, C., Vedaldi, A.: Unsupervised part discovery from contrastive reconstruction, NeurIPS 2021.
>
>
> #### Q4: The improvement on downstream tasks is marginal as shown in Table 6 and Table 7.
>
> A4: 1) Our model can bring **consistent improvements** over various vision tasks, e.g., 3 downstream classification tasks (Table 6) and 3 semantic segmentation tasks (Table 7 and the added comparisons in the following table). 2) As shown in previous work, such improvements on each dataset are convincing to **show the effectiveness** of the proposed model (e.g., for classification, iBoT outperforms BeiT by 0.1% accuracy on Cars dataset, and for segmentation, MAE outperforms BeiT by 0.3 mIOU on ADE-20K dataset).
>
>
> |  model| cityscapes  |coco_stuff10k  |
> |  ----  | ----  |----  |
> | MAE  | 80.30  |38.89 |
> | SemMAE(ours)  | **80.81** |**39.05**  |

---

> > ### Author Response · Authors · 2022-08-02
> > **Response to Reviewer KRLr**
> >
> > ####  Q5: In the end of the line 155: convolutional can AdaIN layers --> convolutional and AdaIN layers
> >
> > A5: Thanks for your comments, and we will carefully double-check the grammar and typos.
> >
> >
> > #### Q6: How about the total training cost including the iBOT pre-training, part learning, semantic guide pre-training, compared to the other pre-training methods.
> >
> > A6: Our pipeline can be separated into part learning and semantic-guided MAE, where the training cost of semantic-guided MAE is almost the same as other pre-training methods (i.e., MAE). Using the off-the-shelf iBOT-pretrained model, our part learning only needs 2 GPU hours (0.2% additional cost). If we also consider the cost of pre-training iBOT (ViT-small for part learning), 25% additional GPU hours would be costly compared to other pre-training methods.
> >
> >
> > #### Q7: The proposed method assumes that the images are object-centric. It's true for ImageNet dataset. How about the proposed method on none-object images, e.g. texture images, scene images, and et.al?
> >
> > A7: The majority of image representation learning tasks focus on learning object-centric image representations. Thus currently, related discussions on texture images and scene images are not performed in our work. Thanks for your insightful comments, and we will discuss how to learn the visual analogue of words for non-object scenarios in future work.

---

> > > ### Comment · Area_Chair_NhM3 · 2022-08-08
> > > **Review Reminder: Please respond to authors.**
> > >
> > > Dear Reviewer, please remember to respond to the author's rebuttal. Thank you.

---

> ### Author Response · Authors · 2022-08-08
> **Looking forward to your feedback**
>
> Dear Reviewer KRLr,
>
> We appreciate your time and valuable comments on our paper. Based on your comments, we did our best to answer questions like why we need semantically guided MAE training, etc. We sincerely hope our responses have addressed your concerns and wish to respectfully ask if you can reconsider the recommendation on our submission. If you still have any questions or ambiguations, we will try our best to address them.

---

### Official Review · Reviewer_NYYu · 2022-07-11

**Rating:** 7
**Confidence:** 5
**Soundness:** 3 good
**Presentation:** 3 good
**Contribution:** 3 good

**Summary:**

In the submission, the authors hope to explore a more efficient/reasonable masking strategy for training masked autoencoders (MAEs). Their intuition is that current masking strategies (e.g., random masking) cannot utilize semantic information contained in images, as what have been done in masked language modeling. Therefore, they proposed that masking semantic parts in the image would help training MAEs, named Semantic-Guided Masking. Based on this idea, the authors devised (i) a semantic part learning strategy for predicting semantic parts from the image; (ii) a MAE masking curriculum called semantic-guided MAE for gradually masking semantic parts in the image. Finally, experiments show the proposed semantic-guided MAE bring significant improvements compared to existing masking strategies.

**Questions:**

1. In Line 147, I am wondering why the authors attempt to remove texture information, and the reason for choosing a large blur kernel.

2. How do the authors decide the part segmentation results of stage-I being good enough? According to my understanding, for a fair comparison, no labels should be introduced.


**Limitations:**

Yes, related limitations have been discussed and there is no obvious negative societal impact.

**Strengths And Weaknesses:**

Strengths:

1. In my opinion, the idea of this paper is interesting. Compared to randomly masking images, masking semantic parts seems reasonable, and experiments show it indeed brings improvements. Meanwhile, the idea of recovering semantic parts of an image giving other parts (or information) is an important research topic of image understanding, this paper should bring some new thoughts to the community.

2. Learning to predict semantic parts from images is a challenging task, especially under the unsupervised training setting. To execute the proposed semantic-guided MAE, the authors proposed a new semantic part learning strategy which is novel and useful. According to my understanding, the performance of following semantic-guided MAE is highly relying on semantic part segmentation results. It is impressive that using only a style-based generator and reconstruction loss can further refine the semantic part information extracted from a pre-trained ViT encoder.

3. This paper is well-written. The motivation and idea of this paper are clear and are well-explained. Although visual transformer (or MAE) made great progress in the past year, it is still chasing language modeling. I am glad to see papers that explore the differences and connections between language tokens and image tokens.

Weakness:

1. The whole pipeline of the proposed method seems a little bit complicated. It first learns to predict part segmentations from a pre-trained iBOT, then retrains an MAE from the scratch. I am curious how about training a random MAE first, and then iteratively performing part learning and semantic-MAE learning. I understand the authors listing it in their future work, yet it still can be discussed in this paper.

2. In table 4, `iBOT-initialized part' performs worse than the baseline model, the authors should have more discussion on it. How will the part learning influence the Semantic-MAE results?

---

> ### Author Response · Authors · 2022-08-02
> **Response to Reviewer NYYu**
>
> Thanks for your valuable comments. We are encouraged that you consider the idea of semantic-guided MAE interesting, reasonable, and well-motivated. We will address each of your questions or concerns below.
>
> #### Q1: The whole pipeline of the proposed method seems a little bit complicated. It first learns to predict part segmentations from a pre-trained iBOT, then retrains an MAE from the scratch. I am curious how about training a random MAE first, and then iteratively performing part learning and semantic-MAE learning. I understand the authors listing it in their future work, yet it still can be discussed in this paper.
>
> A1: Integrating part learning and semantic-MAE learning into one model by iteratively training **is a good idea** to 1) simplify the pipeline and 2) make these two processes mutually reinforce each other. While there are also several **potential challenges**, e.g., *1) a promising initialization:* it may be hard to learn meaningful semantic parts from an early-stage MAE; *2) hyper-parameters:* plenty of experiments are required to decide the optimal iterative steps for each process; *3) convergency:* the training strategy should be well-designed to promise these two processes to reinforce each other instead of harming each other.
>
> In this work, we adopt the idea of divide-and-conquer and separate the pipeline into two aspects. We hope such an exploration can address more researchers' interests in both part learning and part-based representation learning. And in the future, we would like to further conduct a deep discussion on integrating these two processes into one model.
>
>
> #### Q2: In table 4, `iBOT-initialized part' performs worse than the baseline model, the authors should have more discussion on it. How will the part learning influence the Semantic-MAE results?
>
> A2: Precise semantic parts can benefit SemMAE while imprecise parts would decrease the performance. For example, some parts may be mislocated in the background, and when we mask 75% "parts" of an image, the models would be required to predict foreground objects given only background information. Such a task would undermine the representation learning process. As a result, we think it is reasonable that in Table 4 (main submission), the "iBOT-initialized part" performs worse than the baseline model.
>
>
> #### Q3: In Line 147, I am wondering why the authors attempt to remove texture information, and the reason for choosing a large blur kernel.
>
> A3: The motivation for removing texture information is to encourage the attention map to focus on learning spatial/positional information. Using a large blur kernel can 1) effectively remove texture information and 2) improve semantic consistency by encouraging adjacent pixels (of the attention map) to share the same semantic.
>
>
> #### Q4: How do the authors decide the part segmentation results of stage-I being good enough? According to my understanding, for a fair comparison, no labels should be introduced.
>
> A4: There are no part labels that are available to evaluate the part segmentation results. We analyze the part learning process from two aspects, i.e., 1) qualitative analyses as shown in Figure 4 (main submission), and 2) indirect quantitative evaluation of linear probing as shown in Table 4 (main submission).
> It is hard to define "good enough", but it is clear that better part segmentation results are more beneficial to Sem-MAE.

---

> ### Comment · Reviewer_NYYu · 2022-08-09
> **Response to Authors**
>
> Thanks for the authors' reply.
>
> I would keep my original rating as accept (7)
>
>
> Best,

---

### Meta-Review · Area_Chair_NhM3 · 2022-08-22

**Recommendation:** Accept
**Confidence:** Certain

**Metareview:**

Authors present a method attempting to perform Masked Auto-Encoding (MAE) using semantic knowledge, to try to better approximate the semantic MAE seen in language domain. To do this, they leverage an iBOT framework and add some embeddings of the class token to create "part tokens", which are then compared to patch tokens from iBOT to produce attention maps. The objective for this process is a StyleGAN-based image reconstruction.

Once the part attention maps training is done, the network is then used to guide semantic based part masking based on the generated attention maps, for semantic MAE.

SemMAE pretrained networks are then compared against other forms of SSL pretraining on ImageNet 1k, iNa, CUB, Cars, and ADE-20K, demonstrating improvements in all domains.

Pros:
- [R] Idea is interesting / novel
- [R] Well written
- [AC/R] Results improve over baselines


Cons:
- [AC/R] Pipeline is complicated.
- [R] What about starting from random MAE and then adapting based on parts knowledge? Authors respond that this is future work.
- [AC/R] Not convinced parts are visual analog of words. Authors provide benchmark improvements in performance, and qualitative visualization of the parts. However, there is no quantitative assessment of the parts and whether they have true semantic meaning.
- [AC/R] Some improvements are marginal. Authors respond that although marginal in some cases, they are consistent.
- [R] Paper does not discuss more how to deal with background. Authors respond that this is future work.

Overall, all reviewers have changed their assessments to accept, including the one reject reviewer. AC recommends accept, though would be preferable if quantitative assessment of the quality of the semantic parts (for example, by segmentation masks) could be provided.

AC Rating: Accept

**Award:**

No

---

### Decision · Program_Chairs · 2022-09-14

Accept